

# Similarity analysis of turbulent transport and dissipation for momentum, temperature, moisture, and $CO_2$ during BLLAST

João A. Hackerott[1], Mostafa Bakhday Paskyabi[2], Stephan T. Kral[2], Joachim Reuder[2], Amauri P. de Oliveira[1], Edson P. Marques Filho[3], Michel d. S. Mesquita[4], and Ricardo de Camargo[1]

[1]Institute of Astronomy, Geophysics and Atmospheric Sciences, University of Sao Paulo, Sao Paulo, Brazil
[2]Geophysical Institute, University of Bergen, and Bjerknes Centre for Climate Research, Bergen, Norway
[3]Institute of Physics, Federal University of Bahia, Salvador, Brazil
[4]Uni Research Climate and Bjerknes Centre for Climate Research, Bergen, Norway

*Correspondence to:* J. A. Hackerott (joao.hackerott@gmail.com)

**Abstract.** The budget equation components for turbulent kinetic energy (TKE) and the variances of virtual potential temperature, specific humidity, and specific $CO_2$ content have been estimated using the Inertial Dissipation and Eddy Covariance methods. A discussion with four examples is provided about the normalization used for comparing different tracer spectra, divided by the respective char-

acteristic scale squared. A total of 124 high frequency sample segments of a 30-min period from 20 days of the Boundary Layer Afternoon and Sunset Turbulence field campaign were used in order to provide parameterizations for the dimensionless dissipation and residual (i.e. total transport) components as a function of the Atmospheric Surface Layer (ASL) stability parameter, $\zeta$. The results show a similar linear relation for all tracers variance dissipation components, $\Phi_\chi^D \cong 0.4 + 0.2\zeta$,

during the convective ASL, i.e. $-1 < \zeta < -0.1$. Although parameterizations were also proposed for the dimensionless dissipation rate of TKE and tracer variances during stable ASL, we conclude that in this regime, other mechanisms in addition to $\zeta$ may be significantly important. In the stable and near-neutral ASL stability regimes, the transport component for different tracers may not be considered the same. In these conditions, the dissipation component of TKE and tracer variances can have

the same magnitude as the other terms in their respective budget equation.

## 1  Introduction

The Atmospheric Surface Layer (ASL) is characterized by different turbulent flux mechanisms involving momentum and meteorological tracers (Zilitinkevich and Calanca, 2000), such as temperature, moisture and $CO_2$ concentration. The ASL fluxes can be estimated from their respective vari-

ance budget equations (which provide a bridge between the total variance storage and the effects



of variance production, transport, and dissipation) depending on the atmospheric stability (Norman et al., 2012). Therefore, the detailed knowledge of turbulence and its fluxes is crucial in comprehensive understanding of the interactions among various processes between atmospheric physical dynamics and canopy chemical reactions (Foken et al., 2006; Ruppert et al., 2006; Sorbjan, 2009).

Several methodologies for quantifying the dimensionless components of the variance budget equations have been explored in the past decades, mainly for the momentum and heat (temperature and moisture) budgets (Kaimal et al., 1972; Fairall and Larsen, 1986; Edson et al., 1991; Edson and Fairall, 1998; Hartogensis and Bruin, 2005). Fewer studies, however, have considered the respective budgets of some atmospheric tracers, such as specific $CO_2$ content (Ohtaki, 1985; Sahlée et al.,

2008; Sørensen and Larsen, 2010; Sørensen et al., 2014). Nonetheless, little is known about parameterizations for the dissipation and transport components, especially for the $CO_2$ concentration (Iwata et al., 2005) and the stable ASL regime (Pahlow et al., 2001).

Despite the difficulty of providing expensive fast-response sensors for all different fluxes in several height levels, the most preferred technique to estimate the budget components from one level mea-

surements is a combination of two methods: the Eddy Covariance (EC), e.g. Aubinet et al. (2012), and the Inertial Dissipation (ID), e.g. Bluteau et al. (2011). These methods provide information about the variance dissipation and the ASL's characteristic scales by assuming the validity of the Monin-Obukhov Similarity Theory (MOST) and the Kolmogorov power law.

In this context, the height-dependent components in the normalized variance budget equations, such as variance production and transport terms, need to be parameterized, usually as a function

of the atmospheric stability. While the parameterizations for the mechanical and thermodynamical production terms are well-established for different budget equations (Högström, 1996; Foken, 2006; Wyngaard and Coté, 1971), little is known about the transport components due to inherent difficulties of measuring the pressure and the third-order moments of other variables. Thus, transport terms are

usually neglected (Fairall and Larsen, 1986; Hartogensis and Bruin, 2005; Bumke et al., 2014) or treated as a residual term in the budget equations (Yelland and Taylor, 1996; Dupuis et al., 1997; Nilsson et al., 2015).

In this study, we investigate the functional relationship between the dissipation and residual components of the variance budget equations of momentum, heat, moisture, and $CO_2$ concentration,

discussing the proposed parameterizations as a function of the ASL stability. Our analysis is based on 20 days of high frequency observations during the Boundary-Layer Late Afternoon and Sunset Turbulence (BLLAST) field experiment (Lothon et al., 2014), taken at the small-scale heterogeneity station. Considerations about how the results and the proposed methods are useful for different applications, such as studies involving air-sea interactions and canopy displacement length, are also

discussed.

This paper is structured as follows. In Sect. 2, we give a brief introduction on the required preprocess and quality control procedures. Section 3 formulates the budget components and the neces-





sary background about the EC and the ID methods. Section 4 evaluates and describes the results. The final section summarizes with conclusions and a brief discussion about two important applications
of the presented methodologies and results.

## 2   Description of the dataset

### 2.1   Measurement site and data

Measurements of wind velocity components, air temperature and the concentrations of water vapor and carbon dioxide were carried out on a $2\,\mathrm{m}$ micrometeorological mast mounted at the Surface Site
3 (SS3) in Site 1 from the BLLAST experiment (Lothon et al., 2014). The campaign took place in France (*Plateau de Lannemezan*), a few kilometers North of the foothills of the Pyrenees, from 14$^{\text{th}}$ June to 8$^{\text{th}}$ July, 2011. The area around SS3 is characterized by a flat surface ($150 \times 150\,\mathrm{m}^2$) covered mainly by short grass. The measuring system for sampling frequency of 20 Hz includes a sonic anemometer CSAT3 for collecting data of temperature and wind velocity, a gas analyzer LICOR
7500 for monitoring the concentrations of $H_2O$ and $CO_2$, and a Paroscientific microbarometer for measuring the air pressure, as illustrated in Fig. 1.

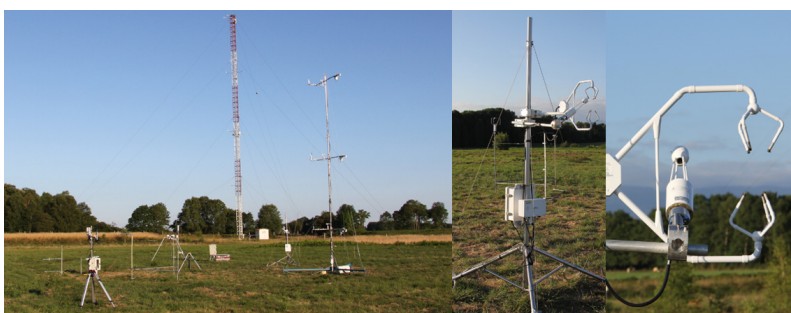

**Figure 1.** Field setup (left) of Surface Site 3 in Site 1 of the BLLAST campaign, micrometeorological mast (middle), and detail of the sonic anemometer CSAT3 and the gas analyzer LICOR 7500 (right).

The temperature was converted to the virtual potential temperature using the moisture and pressure data. From now on, the temperature or the potential temperature ($\theta$) will be referred to as virtual potential temperature. Considering the pressure and the air temperature, and neglecting the moisture
effect on the air density, the $H_2O$ and $CO_2$ concentrations were converted to the specific humidity ($q$) and specific $CO_2$ content ($c$), in $[\mathrm{mg\,kg^{-1}}]$. Additionally, the wind data was rotated into the streamwise wind (i.e. $\overline{v} = \overline{w} = 0$), following Lee et al. (2005), for every 15-min segment, which is the interval length applied for the ID method.





## 2.2 Quality assessment of the dataset

Based on solar radiation data (Lothon et al., 2014), 20 days with low cloud coverage have been selected for this study. Thus, the $14^{th}$, $15^{th}$, $24^{th}$ and $30^{th}$ days of June from the BLLAST campaign have not been considered.

In order to reduce the effect of flow distortion induced by the probes and the mast, the dataset was limited to the undistorted sector by removing the wind data that comes from a sector of $\pm 30°$ in the
foremast direction (Li et al., 2013).

The data also underwent a conventional spike removal algorithm (Foken et al., 2005). It is considered for this study only 30-min segments with more than $95\%$ of the original length preserved.

The EC and ID methods are based on the assumption of stationarity. The steady state test was applied for the three wind velocity components by dividing every 30-min segment into four equal
intervals and comparing the variances among them. If the variances deviate more than $50\%$ compared to the lowest variance within the segment, the whole segment is discarded.

The so-called Ogive test for $\overline{u'w'}$ (Foken et al., 2006) was applied as an additional quality test, where $u'$ and $w'$ are the horizontal and vertical velocity fluctuation components. This test is used to describe the distribution of the covariances across the frequency domain, verifying if the segment
encloses all the turbulent eddies that contribute to the overall flux. The ogive curves, $Og_{u'w'}$, are defined by taking the cumulative integral of the turbulent flux co-spectrum ($Co_{u'w'}$), starting with the highest frequency:

$$Og_{u'w'}(n) = \int_{\infty}^{n} Co_{u'w'}(n)dn. \tag{1}$$

The test compares the $Og_{u'w'}$ for two distinct frequencies, $n$, and evaluates the rate $R_{Og}$, defined in Eq. (2). In this study, the 0.5-hour (0.5h) segments were expanded one hour back and one hour forward, in order to have a larger segment of 1.5-hour (1.5h). If $R_{Og}$ was greater than the threshold $90\%$, it was assumed that the ogive converged and the original segment is large enough to satisfy that low frequency parts are included in the measured turbulent flux. Here, $81\%$ of the total available
segments were accepted after this test.

$$R_{Og} = \left| \frac{Og(n_{0.5h})}{max[Og(n_{1.5h})]} \right|. \tag{2}$$

The validity of the Taylor Hypothesis was examined considering only segments where the standard deviation of the streamwise wind, in isotropic turbulent flows, was smaller than half of the mean wind
velocity (Willis and Deardorff, 1976).



A few segments were rejected after visual inspection of all automatically qualified datasets. A total of 124 segments of 30-min period are labeled as accepted and used in the present analysis. The data contains most of the whole 24-hour day period, with gaps on the hours 11.25, 15.25, 18.25 and 19.25 UTC, when no segment was accepted. The averages and standard deviations of the observed

wind magnitude, $\theta$, $q$, and $c$ are $2.1 \pm 0.9 \, \mathrm{m \, s^{-1}}$, $295 \pm 5 \, \mathrm{K}$, $14 \pm 3 \, \mathrm{g \, kg^{-1}}$, and $746 \pm 39 \, \mathrm{mg \, kg^{-1}}$, respectively.

## 3 Budget of the scalar variances

### 3.1 The budget equations of the variances

The variance budget equations for the momentum (commonly referred to as Turbulent Kinetic En-

ergy, TKE) and the different scalars $\chi_i$ (i.e. $\chi_1 = \theta$; $\chi_2 = q$; $\chi_3 = c$), for the isotropic homogeneous turbulent flow, in the absence of subsidence, can be expressed as follows:

$$\frac{1}{2}\frac{\partial \overline{u'^2}}{\partial t} = -\overline{u'w'}\frac{\partial \overline{u}}{\partial z} + \frac{g}{\theta_0}\overline{\theta'w'} - \frac{\partial \overline{u'^2 w'}}{\partial z} - \frac{1}{\rho_0}\frac{\partial \overline{p'w'}}{\partial z} - \epsilon, \tag{3}$$

$$\frac{1}{2}\frac{\partial \overline{\chi_i'^2}}{\partial t} = -\overline{\chi_i' w'}\frac{\partial \overline{\chi_i}}{\partial z} - \frac{1}{2}\frac{\partial \overline{\chi_i'^2 w'}}{\partial z} - N_{\chi_i}, \tag{4}$$

where $\epsilon$ is the dissipation rate of TKE, $N_{\chi_i}$ the dissipation rates of the tracer variances, $g$ the gravitational acceleration, $\theta_0$ the reference virtual potential temperature, $\rho_0$ the reference air density, $z$ the height above surface, and $p'$ the atmospheric pressure fluctuation. Overbars and primes are used to denote Reynolds averages and fluctuations from this average, respectively. It is worth mentioning that the effect of roughness length on the observational height $z$ is neglected, since deviations of the

order of a few centimeters in $z$ would induce much smaller uncertainties than other factors involved in the EC and ID methods.

For homogeneous turbulence within the ASL, the MOST asserts that the vertical turbulent fluxes are described by characteristic scales. Rewriting Eqs. (3) and (4) in a dimensionless form, and assuming stationary conditions, gives:

$$\underbrace{\frac{\kappa z}{u_*}\frac{\partial \overline{u}}{\partial z}}_{\Phi^M} - \underbrace{\frac{\kappa z \theta_*}{u_*^2}\frac{g}{\theta_0}}_{\zeta} - \underbrace{\left[\frac{\kappa z}{u_*^3}\frac{\partial \overline{u'^2 w'}}{\partial z} + \frac{\kappa z}{u_*^3}\frac{1}{\rho_0}\frac{\partial \overline{p'w'}}{\partial z}\right]}_{\Phi_u^T} - \underbrace{\frac{\kappa z}{u_*^3}\epsilon}_{\Phi_u^D} = 0, \tag{5}$$

$$\underbrace{\frac{\kappa z}{\chi_{i_*}}\frac{\partial \overline{\chi_i}}{\partial z}}_{\Phi_{\chi_i}^H} - \underbrace{\frac{\kappa z}{2 u_* \chi_{i_*}^2}\frac{\partial \overline{\chi_i'^2 w'}}{\partial z}}_{\Phi_{\chi_i}^T} - \underbrace{\frac{\kappa z}{u_* \chi_{i_*}^2}N_{\chi_i}}_{\Phi_{\chi_i}^D} = 0, \tag{6}$$





where $\kappa$ is the von Kármán constant ($\kappa \approx 0.4$), $\Phi^M$ the dimensionless mechanical production of TKE, and $\Phi^H_{\chi_i}$ the dimensionless thermodynamic production of $\chi_i$ variance. The superscript $T$ indicates the transport and $D$ the dissipation dimensionless components, and the subscript $*$ is a reference to the ASL characteristic scale. The buoyancy component ($\zeta$) is, by definition, the Monin-Obukhov stability parameter, with negative values indicating convective and positive values indicating stable conditions within the ASL. Furthermore, according to MOST, $\Phi^M$ and $\Phi^H_{\chi_i}$ are universal functions, which depend only on $\zeta$. Finally, the vertical transport of TKE and the divergence of pressure transport in Eq. (5) are treated as the total transport of TKE, $\Phi^T_u$.

There are several parameterizations available in the literature for $\Phi^M$ and $\Phi^H_{\chi_i}$. According to the review of Foken (2006), the most acceptable universal functions for momentum, sensible heat and water vapor are described by Högström (1996), valid for $-2 < \zeta < 0.5$. Therefore, we parameterize the mechanical and thermodynamical production terms as follows:

$$\Phi^M = \begin{cases} (1-19\zeta)^{-\frac{1}{4}}, & \zeta \leq 0 \\ 1+5.3\zeta, & \zeta > 0 \end{cases} \tag{7}$$

$$\Phi^H = \begin{cases} (0.95(1-11.6\zeta)^{-\frac{1}{2}}, & \zeta \leq 0 \\ 1+8\zeta, & \zeta > 0 \end{cases} \tag{8}$$

Considering that Eq. (8) is invariant to the tracer type (i.e. $\Phi^H = \Phi^H_{\chi_i}$), also suggested by other authors, e.g. Dyer (1974); Hill (1989), an equation for the dissipation rates of different tracer variances can be expressed as:

$$\frac{N_{\chi_i}}{\chi_{i*}^2} = \frac{u_*}{\kappa z}\left(\Phi^H - \Phi^T_{\chi_i}\right). \tag{9}$$

Considering the spectral theory (explained in Sect. 3.3), Eq. (9) suggests that the usual spectra normalization (Sahlée et al., 2008), power spectra divided by the squared characteristic scale, originally proposed by Kaimal et al. (1972), makes the spectra collapse within the inertial subrange only if the transport term is also invariant to the tracer type. Analogously, this fact can be also observed for the TKE power spectrum in neutral conditions ($\zeta = 0$), when $\Phi^M \approx \Phi^H$.

The dissipation components are usually parameterized as functions of $\zeta$, as summarized by Hill (1997). It is consensus to distinguish the parameterization for different stability regimes, however, the proposed functions vary considerably in their general shape, the corresponding coefficients, and the stability range they are defined for (e.g. Wyngaard and Coté, 1971; Ohtaki, 1985; Kader, 1992; Hartogensis and Bruin, 2005; Pahlow et al., 2001). Based on the present results, it is assumed linear function for $\Phi^D(\zeta)$ in the different stability regimes. In the special case of near-neutral conditions, $\Phi^D$ is balanced by $[1 - \Phi^T]$, which might not be a constant like the mechanical production component. Beyond that, from Eq. (6), one can follow $\lim_{\zeta \to 0} \Phi^D_\theta = \infty$, as it was observed by Kader (1992), which will be discussed in Sect. 3.2. Thus, the dissipation component analysis is limited to $|\zeta| > 0.1$.





Since $\Phi^T$ is regarded as a residual term in Eqs. (5) and (6), its parameterization is then defined in this study as:

$$\Phi_u^T = \Phi^M - \Phi_u^D - \zeta \,, \quad |\zeta| > 0.1 \tag{10}$$


$$\Phi_{\chi_i}^T = \Phi^H - \Phi_{\chi_i}^D \,, \quad |\zeta| > 0.1 \tag{11}$$

### 3.2 The ASL characteristic scale

The EC method computes the second-order moments directly from a high-frequency dataset. According to MOST, the ASL characteristic scale for wind velocity (i.e. friction velocity) and tracers are defined in terms of these second-order moments as follows:

$$u_* = \left[ \left(\overline{u'w'}\right)^2 + \left(\overline{v'w'}\right)^2 \right]^{\frac{1}{4}}, \tag{12}$$

$$\chi_{i_*} = -\frac{\overline{\chi_i'w'}}{u_*}. \tag{13}$$


Regardless of the simplicity of these equations, special care must be taken in order to guarantee two main hypotheses, i.e. the stationarity of the flow, and the representation of all relevant scales of turbulent eddies in their respective covariances. While the first suggests short period segments, the second suggests longer period segments. In order to meet both criteria in the best possible way, we decided to use 30-min segments to estimate the characteristic scale values. The aforementioned quality control and assurance procedure satisfied both hypotheses reasonably.

It is noteworthy that, in the absence of a tracer flux, the characteristic scale values in Eq. (13) vanish (i.e. $\chi_{i_*} = 0$), and then all terms in Eq. (6) become undefined. Nevertheless, observations indicate that, in neutral conditions, $\Phi^M$ and $\Phi^H$ asymptotically approach a constant (Dyer, 1974; Högström, 1996). Also, $\theta_* \to 0$ only when $\zeta \to 0$, and then the singularity occurs just during neutral ASL for the equations involving temperature. However, this is not the case for $q_*$ and $c_*$, for which the sign is asserted by the corresponding vertical gradient direction, that is not necessarily correlated to $\zeta$. Then, in order to evaluate the behavior of universal functions against $\zeta$, the segments with $|q|_* < 0.01 \, \mathrm{g\,kg^{-1}}$ and $|c|_* < 0.1 \, \mathrm{mg\,kg^{-1}}$ were excluded. The Taylor Hypothesis test performed in the quality control is sufficient to guarantee $u_* \not\approx 0$, avoiding singularities in Eqs. (13), (5) and (6).

### 3.3 Spectral theory

According to the Kolmogorov power law, there is a range of wave numbers, $k$, within the momentum energy spectrum, $S_u$, called inertial subrange, where the energy is transferred from large scale eddies



to small scale ones, at a constant dissipation rate of TKE, $\epsilon$. Within this inertial subrange the energy

production and dissipation, are sufficiently smaller than the spectral density of the streamwise wind velocity, which then depends only on $k$ and $\epsilon$. This theory, originally formulated by Kolmogorov (1941) for the momentum variance spectrum, was later implemented for heat (Corrsin, 1951), moisture (Edson et al., 1991), and specific $CO_2$ content (Sørensen and Larsen, 2010; Sahlée et al., 2008).

Considering the Taylor Hypothesis of "frozen" turbulence, it is possible to express $S$ in terms

of frequency, $n$, instead of $k$. Therefore, within the inertial subrange, the spectral densities may be expressed in terms of frequency power spectrum of longitudinal wind velocity, $S_u(n)$, and tracers, $S_{\chi_i}(n)$:

$$S_u(n) = \alpha \epsilon^{\frac{2}{3}} n^{-\frac{5}{3}} \left( \frac{2\pi}{U} \right)^{-\frac{2}{3}}, \tag{14}$$

$$S_{\chi_i}(n) = \beta \epsilon^{-\frac{1}{3}} n^{-\frac{5}{3}} N_{\chi_i} \left( \frac{2\pi}{U} \right)^{-\frac{2}{3}}, \tag{15}$$

where $U$ is the mean longitudinal wind velocity, $\alpha = 0.52$ is the Kolmogorov constant, and $\beta = 0.8$ is the Obukhov-Corrsin constant, following Högström (1996). It is assumed in this study that $\beta$ is invariant to the tracers type (Hill, 1989; Iwata et al., 2005). As a consequence of isotropy, Eq. (14) can also be expressed for the vertical and transversal wind components, $w$ and $v$, using $\alpha_v = \alpha_w = \frac{4}{3}\alpha_u$

(Kaimal et al., 1972).

The spectral densities are computed using the Fast Fourier Transform (FFT). This technique, however, requires additional data treatment in order to reduce undesirable inherent noise (Stull, 1988). Thus, the segments were detrended and 5% tapered by the Tuckey Window function (Bloomfield, 2000).

The segment length is crucial for the ID method. Based on visual inspection of the spectral signals, we decided to use segments of 18,000 data samples (15-min), which is large enough to ensure that the low-frequency limit of the inertial subrange is well resolved in the FFT result. The ID method is very sensitive to the stationarity assumption (Bluteau et al., 2011), which is more likely to be violated when using longer data segments (e.g. 30-min segment) and thus causing elevated values

of the spectral density due to the white noise.

After providing $S_u(n)$, $\epsilon$ is estimated from Eq. (14) using a nonlinear fit within the inertial subrange. $N_{\chi_i}$ is then estimated, also by applying a nonlinear fit, based on Eq. (15), to the respective inertial subrange of $S_{\chi_i}(n)$. The inertial subrange interval is estimated using the inertial subrange detector method, described in Appendix A.

In order to combine the results from the EC and ID methods, which are estimated for different segment lengths, it is necessary to convert the dissipation rates calculated for the 15-min segment to the 30-min segment. For this, we propose an iterative methodology. First, the 30-min intervals are enlarged 14-min back and forward, resulting in 69,600 data samples. Within the iterative process,



the dissipation rates for a 15-min interval are calculated, starting at the beginning of the extended

segment. This procedure is then repeated by moving the start point with a time step of 1-min until

the end of the extended segment is reached. This iteration ensures that almost all data of the original

30-min interval (excluding the first and last 1 min period) are weighted equally. The dissipation rate

for the 30-min segment is then defined as the median of the 44 iterations. We use median instead

of the mean to prevent the influence of eventual overestimated dissipation rates originated from

non-stationarities.

## 4   Results

Following the premise that the variance budget components may be expressed as a function of $\zeta$, the

results in this section are classified into three ASL stability regimes, convective ($\zeta < -0.05$), near-

neutral ($|\zeta| \leq 0.05$), and stable ($\zeta > 0.05$), distinguished by three different colours in the presented

figures.

Our results will be discussed in three parts: first a discussion of MOST parameters and dissipation

rates derived by the EC and ID methods; second the examination of the dissipation and residual

components of the variance budget equation; and finally a discussion of four selected examples.

### 4.1   MOST parameters and dissipation rates

The time series of the Monin-Obukhov stability parameter and the ASL characteristic scale values

are displayed in Fig. 2, where the colours distinguish the stability regimes, blue for the stable, red

for the convective, and black for the near-neutral. The 124 available segments covered in a similar

quantity all the three regimes, 35% for the convective, 27% for the stable, and 38% the near-neutral

regime. The $\zeta$ ranges from -0.99 to 1.06, with five occurrences with $\zeta \geq 0.5$, which are out of the

validity range of Eqs. (7) and (8).

The friction velocity, $u_*$, significantly oscillates depending on the ASL stability regime. It is

observed $u_* = 0.22 \pm 0.07\ \mathrm{m\,s^{-1}}$ for the segments collected during the convective ASL, while for

the stable regime, $u_* = 0.12 \pm 0.04\ \mathrm{m\,s^{-1}}$, excluding the observations done during the night time

between June 26[th] and 27[th]. The segments measured during this night registered an elevated mean

wind velocity, $3.4 \pm 0.2\ \mathrm{m\,s^{-1}}$, and consequently high values for $u_*$, $0.280 \pm 0.016\ \mathrm{m\,s^{-1}}$. In general,

during the convective ASL regimes, the wind velocity was also higher ($2.1 \pm 0.9\ \mathrm{m\,s^{-1}}$) than during

the stable ASL, when it was registered a mean wind velocity of $1.4 \pm 0.6\ \mathrm{m\,s^{-1}}$, excluding the

referred night.

The stability regimes are visually correlated to the characteristic scale values, $\theta_*$, $q_*$, and $c_*$. From

the definition of $\zeta$, it is assigned to the same sign as $\theta_*$, however, for $q_*$ and $c_*$, the flux direction

is less dependent on the stability. During the night time (most likely stable regime), due to the





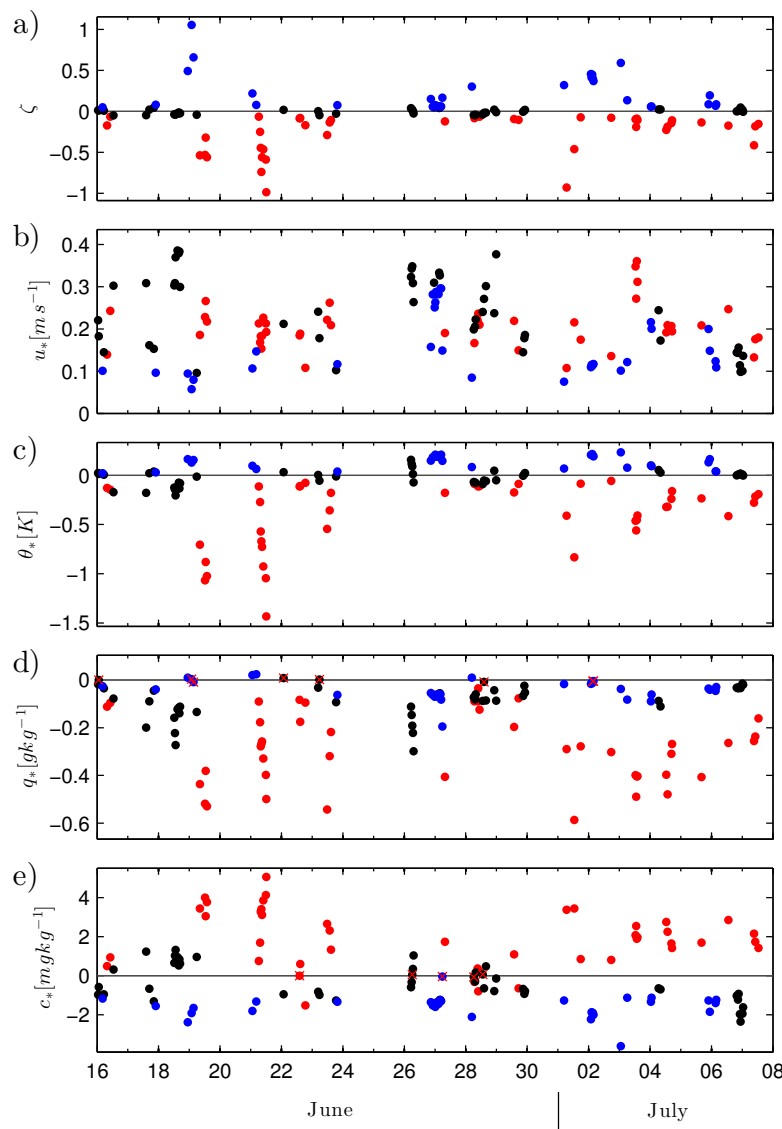

**Figure 2.** Times series, estimated for 30-min segments collected during the 2011 BLLAST campaign, of: (a) Monin-Obukhov stability parameter ($\zeta$); (b) friction velocity [$\mathrm{m\,s^{-1}}$]; (c) characteristic scales of virtual potential temperature [K]; and (d) characteristic scales of specific humidity [$\mathrm{g\,kg^{-1}}$]; and (e) characteristic scales of specific $CO_2$ content [$\mathrm{g\,kg^{-1}}$]. The colours indicate the ASL stability regimes, convective (red), near-neutral (black), and stable (blue). The red crosses on (d) and (e) indicate the data that are not considered for the calculation of the dissipation and residual components of variance budget equations.





lower temperature, the relative humidity is higher, and consequently the vertical moisture gradient is weaker, considering a near saturated soil. In this condition, however, the ASL is still dryer than the
ground below, which provides an upward moisture flux ($q_* < 0$) also for the stable regimes.

The time series of $c_*$ is also correlated to $\zeta$ and $\theta_* > 0$, however, as observed for $q_*$, the sign of $c_*$ does not necessarily follow the sign of $\zeta$, as it is observed on the days 22nd, 28th, and 29th of June. The sign of $c_*$ follows the photosynthesis activity, being positive usually during the daytime. The magnitude of all ASL characteristic scale values are affected by the ASL turbulence, increasing when
the turbulent eddies are stronger, either because of mechanical or buoyant turbulence production.

The seven segments of $q_*$ and five segments of $c_*$, marked with a red cross in Fig. 2 are rejected from the analysis of variance budget components, since they result on singularities in Eq. (6), regardless of the ASL stability regime.

The ID method is used in the calculation of the dissipation rates. The spectra of all segments were
visually analyzed for the three wind components and we decided for using the vertical velocity in Eq. (14), due to the more stationary state of this component when compared to the horizontal ones. This observation was also commented by Nilsson et al. (2015) in their study for the afternoon transition of TKE budget, using the BLLAST dataset, but from a different micrometeorological tower.

Figure 3 summarizes the dissipation rates for the BLLAST datasets that could be estimated. Sim-
ilar to what is observed for $u_*$, the dissipation rate of TKE is more sensitive to the wind magnitude. We highlight the afternoon of June 18th and the morning of June 26th, when although the stability is labeled as near-neutral, it was registered relatively strong winds and, consequently, higher values of $u_*$ and $\epsilon$ are observed.

Despite the fact that the dissipation rates of $\theta$, $q$, and $c$ variances are related to $\epsilon$, in Eq. (15), the magnitude of the dissipation rates of these tracers are more sensitive to their respective fluxes than the wind variation. This fact is well observed on the segments collected during June 21st, July 1st and 3rd, when the intensity of the dissipation rates follow almost the same pattern observed for the characteristic scale values. Thus, the $N_\theta$, $N_q$, and $N_c$ are also correlated with $\zeta$, mainly for the
convective period (day time) as it was explained before for the ASL characteristic scales.

### 4.2 Dissipation and residual components in the variance budget equation

The dimensionless dissipation rates, $\Phi^D$, are presented in Fig. 4 with respect to the different stability regimes, convective (red), near-neutral (black), and stable (blue). Using the BLLAST data, we fit the dissipation rates by the following $\zeta-$dependent linear relationship:

$$\Phi^D = \zeta \, a^\pm + b^\pm, \tag{16}$$

where the superscripts indicate convective ($-$) and stable ($+$) regimes, and $a^\pm$ and $b^\pm$ are the slope and intercept coefficients for the linear regression, respectively.





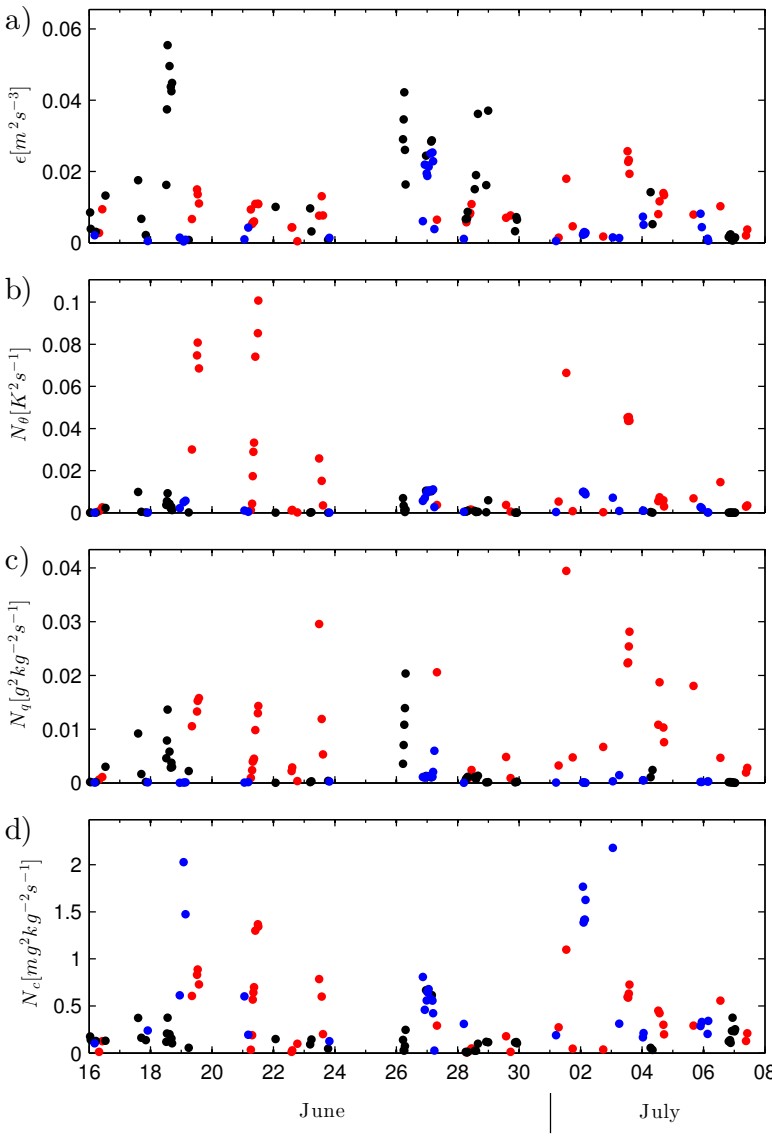

**Figure 3.** Times series, estimated for 30-min segments collected during the 2011 BLLAST campaign, of the dissipation rates of: (a) TKE [$m^2\,s^{-3}$]; (b) the virtual potential temperature variance [$K^2\,s^{-1}$]; (c) the specific humidity variance [$g^2\,kg^{-2}\,s^{-1}$]; and (d) the specific $CO_2$ content variance [$g^2\,kg^{-2}\,s^{-1}$]. The colours indicate the ASL stability regimes, convective (red), near-neutral (black), and stable (blue).





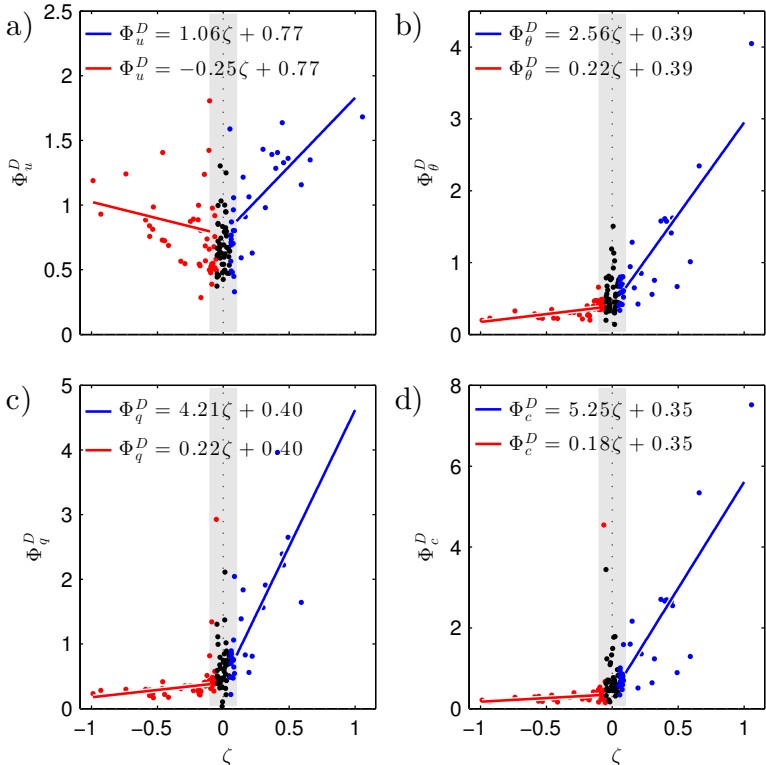

**Figure 4.** Scatterplot of the dimensionless dissipation component ($\Phi^D$) against the Monin-Obukhov stability parameter ($\zeta$) for the budget equation of: (a) TKE; (b) variance of virtual potential temperature ($\theta$); (c) variance of specific humidity ($q$); and (d) variance of specific $CO_2$ content ($c$). The colours indicate the ASL stability regimes, convective (red), near-neutral (black), and stable (blue). The red and blue lines are the linear regressions estimated for the convective and stable regime, respectively. The gray area corresponds to $|\zeta| \leq 0.1$, where the data is not considered for the linear fit.





The scatterplots of the $\Phi^D$ against $\zeta$ show that it is difficult to find an appropriate functional
form for the stable regimes due to the high scattering (i.e. relatively high Root Mean Square Error,
$RMSE$) and the small values for the correlation coefficients of the fitting curves (i.e. $R^2 < 0.7$).
Thus, we held the same intercept coefficient for the stable regime as the one estimated for the con-
vective regime, resulting in a decreased degree of freedom for the linear regression on the stable
ASL datasets. Furthermore, it is evident on the results for $\Phi_q^D$ and $\Phi_c^D$ that the range defined for
near-neutral conditions ($|\zeta| \leq 0.05$) is not enough to avoid the effect of singularities, described on
Sect. 3.1 and 3.2. Therefore, we decided to enlarged this range for $|\zeta| \leq 0.1$ (gray shaded area in
Fig. 4) in order to provide a less biased parameterization. The slope and intercept coefficients are
summarized in Table 1, together with their respective $R^2$ and $RMSE$.

**Table 1.** Coefficients $a^\pm$ and $b^\pm$ for regression defined in Eq. (16). $R^2$ and $RMSE$ are the squared correlation
coefficient and the root mean square error, respectively.

|  |  | $\Phi_u^D$ | $\Phi_\theta^D$ | $\Phi_q^D$ | $\Phi_c^D$ |
|---|---|---|---|---|---|
| Convective | $b^-$ | 0.77 | 0.39 | 0.40 | 0.356 |
|  | $a^-$ | -0.25 | 0.22 | 0.22 | 0.18 |
|  | $R^2$ | 0.04 | 0.28 | 0.19 | 0.28 |
|  | $RMSE$ | 0.31 | 0.09 | 0.11 | 0.07 |
| Stable | $b^+$ | 0.77 | 0.39 | 0.40 | 0.356 |
|  | $a^+$ | 1.06 | 2.56 | 4.21 | 5.26 |
|  | $R^2$ | 0.42 | 0.66 | 0.32 | 0.62 |
|  | $RMSE$ | 0.23 | 0.50 | 0.74 | 1.1 |

Although $R^2$ is lower for the convective ASL regimes, the $RMSE$ is much lower in this regime
compared to the stable ASL, indicating more precision of the regression curves during the convec-
tive regime. The results for $\Phi_u^D$ show a weak relation with $\zeta$ for both the convective and the stable
regimes. As mentioned in Sect. 4.1, the dissipation rate of TKE is sensitive to the wind velocity,
which therefore is more affected by the mesoscale atmospheric conditions than the local turbulence
characteristics. While there are several possible factors that influence the quality of the $\Phi_u^D$ measure-
ments, we conclude that its parameterization should include the effect of mean wind velocity as also
recommended by Yelland and Taylor (1996). They proposed to divide the observed data into three
wind regimes: $3 - 7$, $10 - 12$ and $17 - 20\ \mathrm{m\,s}^{-1}$, respectively. However, the BLLAST datasets fall
only within the first recommended range, which emphasizes the need for considering more relevant
mechanisms rather than only $\zeta$.




The high scattering on stable ASL regime has also been reported by other studies (e.g., Pahlow et al., 2001; Hartogensis and Bruin, 2005; Nilsson et al., 2015), and as a consequence, the available parameterizations vary substantially in the literature. For example, Pahlow et al. (2001) found $a^+ = 5$ and $b^+ = 0.61$, while Hartogensis and Bruin (2005) found $a^+ = 2.5$ and $b^+ = 8$ for $\Phi_u^D$. The only consensus is that $\Phi_u^D$ increases as the ASL becomes more and more stable. For $\Phi_\chi^D$, however, there is no consensus in the literature and few parameterizations are available. Pahlow et al. (2001) suggests a constant value near to unity in the range $0 < \zeta < 100$, although their data varies randomly between 0.1 and 10. In general, the data presented in Fig. 4 shows an increasing magnitude of $\Phi^D$ on stable ASL and significant differences between the slope coefficients of the parameterizations for different tracers.

The parameterizations for $\Phi_{\chi_i}^D$, within the convective regimes, are approximately identical, i.e. $a^- \cong 0.2$ and $b^- \cong 0.4$. These values are in agreement with the results from Ohtaki (1985) for $c$, and Kader (1992) for $\theta$ and $q$. Although they proposed different function shapes, their curves can be approximated by linear functions in the stability range, $-1 < \zeta \leq -0.2$, with a similar slope and slightly higher magnitudes than those observed from this study. For $-0.2 < \zeta < -0.1$, the parameterizations proposed by Kader (1992) and Ohtaki (1985) cannot be considered linear and increase rapidly in magnitude. This fact may be a consequence of the influence of the singularities that occur in near-neutral regimes for the tracers and were not neglected by these authors. Note that the formulations proposed by Kader (1992) need to be multiplied by the von Kármán constant in order to compare to our results.

The transport components, $\Phi^T$, are estimated as a residual term in the variance budget equations. The analysis is restricted to the stable range for $\zeta < 0.5$ because the parameterization for $\Phi^M$ and $\Phi^H$, Eqs. (7) and (8), are specified for this range. Thus, the proposed parameterization curves, derived from Eqs. (10) and (11), are expressed according to Eqs. (17) and (18), with coefficients $a^\pm$ and $b^\pm$ summarized in Table 2.

$$\Phi_u^T = \begin{cases} (1 - 19\zeta)^{-\frac{1}{4}} + \zeta\, a^- + b^- , & -1 < \zeta < -0.1 \\ b^+ + \zeta\, a^+ , & 0.1 < \zeta < 0.5 \end{cases} \tag{17}$$

$$\Phi_{\chi_i}^T = \begin{cases} 0.95(1 - 11.6\zeta)^{-\frac{1}{2}} + \zeta\, a^- + b^- , & -1 < \zeta < -0.1 \\ b^+ + \zeta\, a^+ , & 0.1 < \zeta < 0.5 \end{cases} \tag{18}$$

The scatterplots for $\Phi_{\chi_i}^T$ against $\zeta$, presented in Fig. 5, indicate the convergence to low values in the convective regimes, showing that these terms can be neglected. On the other hand, it is clear that the importance of the $\Phi^T$, on the stable regime increases as the ASL becomes more stable, when





**Table 2.** Coefficients $a^\pm$ and $b^\pm$ for the regressions defined in Eqs. (17) and (18).

|  |  | $\Phi_u^T$ | $\Phi_\theta^T$ | $\Phi_q^T$ | $\Phi_c^T$ |
|---|---|---|---|---|---|
| Convective | $b^-$ | -0.77 | -0.39 | -0.40 | -0.36 |
|  | $a^-$ | -0.75 | -0.22 | -0.22 | -0.18 |
| Stable | $b^+$ | 0.23 | 0.61 | 0.60 | 0.65 |
|  | $a^+$ | 3.24 | 5.44 | 3.79 | 2.74 |

the estimated magnitudes of $\Phi^T$ are comparable to the other components of the respective budget equation, also for $\Phi_u^T$.

Although high scatter is observed for $\Phi_u^T$, the proposed parameterization for $-1 < \zeta < -0.1$ is in agreement with the results from other studies, such as Dupuis et al. (1997), who proposed a linear curve $\Phi_u^T = -0.65\zeta$ for $-7 < \zeta < 0$, and Nilsson et al. (2015) which the parameterization is approximated for $\Phi_u^T = -0.5\zeta$ when $-8 < \zeta < 0$. The curves proposed by these authors suggest a convergence to low values in near-neutral stability regimes, in agreement with Fairall and Larsen (1986) who suggested that $\Phi_u^T$ is about 25% of $\Phi_u^D$ in this regime. Our results, however, show that the mean ratio between $\Phi_u^T$ and $\Phi_u^D$ for $|\zeta| \leq 0.05$ is $0.6 \pm 0.4$.

### 4.3 Case examples

In this section, four 30-min segments are selected with the following ASL stability characteristics for a detailed analysis:

1. June 21$^{\text{st}}$, $11:45:00$ UTC - Convective ($\zeta = -0.6$);

2. June 27$^{\text{th}}$, $00:15:00$ UTC - Low stable ($\zeta = 0.08$);

3. June 18$^{\text{th}}$, $13:15:00$ UTC - Near-neutral ($\zeta = -0.04$);

4. June 18$^{\text{th}}$, $22:45:00$ UTC - Stable ($\zeta = 0.5$).

The estimated components of the variance budget equations, Eqs. (5) and (6), are summarized in Table 3 together with the respective dissipation rates normalized by the squared characteristic scale.

Although Cases 2 and 3 are in a stability regime away from the validity of the proposed functions (i.e. $|\zeta| \leq 0.1$), the parameterizations for the dissipation rate enclose all cases with exception of Case 4 for $\Phi_\theta^D$ and $\Phi_c^D$, considering an acceptable error of 20%. In Case 4, which is an example of the ASL stable regime, the estimated $\Phi_\theta^D$ and $\Phi_c^D$ are 2.5 and 3.3 times greater than the values expected from the suggested parameterizations.





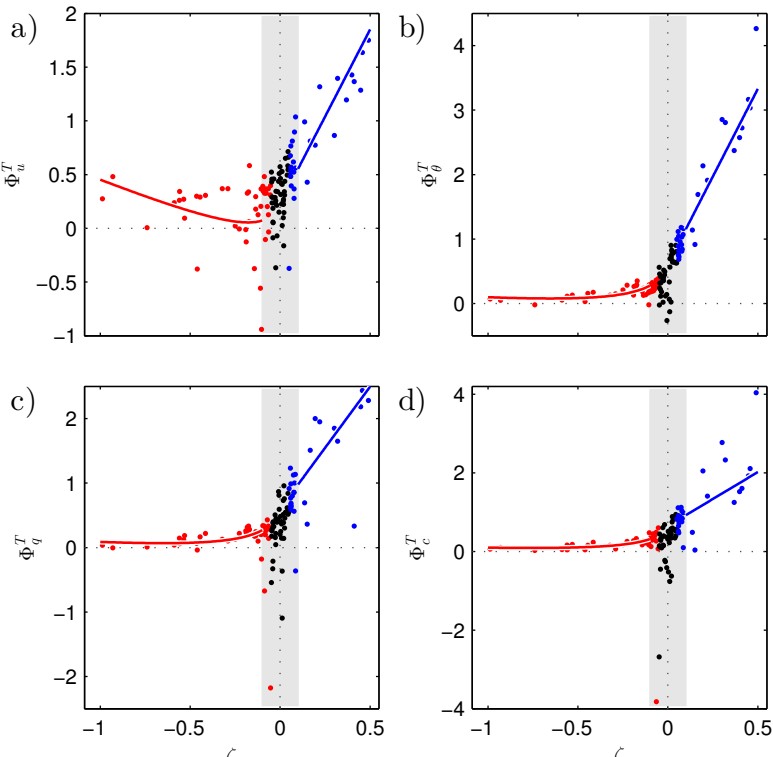

**Figure 5.** Scatterplot of the dimensionless total transport component ($\Phi^T$) against the Monin-Obukhov stability parameter ($\zeta$) for the budget equation of (a) TKE; (b) variance of virtual potential temperature ($\theta$); (c) variance of specific humidity ($q$); and (d) variance of specific $CO_2$ content ($c$). The colours indicate the ASL stability regimes convective (red), near-neutral (black), and stable (blue). The lines are the total transport components estimated as a residual of the variance budget equations, considering the suggested parameterizations for the dimensionless dissipation and dynamical production components. The gray area corresponds to $|\zeta| \leq 0.1$, where the data is not considered for the parameterizations.





**Table 3.** Monin-Obukhov stability parameter ($\zeta$), mechanical ($\Phi^M$) and thermodynamical ($\Phi^H$) production, dissipation ($\Phi^D$), and transport ($\Phi^T$) components of the variance budget equations, and the ratio between the dissipation rates and the corresponding squared characteristic scale in unit of $[\text{s}^{-1}]$, for TKE, virtual potential temperature ($\theta$), specific humidity ($q$), and specific $CO_2$ content ($c$), for the four Cases.

| Case: | 1 | 2 | 3 | 4 |
|---|---|---|---|---|
| $\zeta$ | -0.6 | 0.08 | -0.04 | 0.5 |
| $\Phi^M$ | 0.5 | 1.4 | 0.9 | 3.6 |
| $\Phi^H$ | 0.3 | 1.6 | 0.8 | 4.9 |
| $\Phi^D_u$ | 0.9 | 0.8 | 0.9 | 1.4 |
| $\Phi^D_\theta$ | 0.3 | 0.7 | 0.5 | 0.7 |
| $\Phi^D_q$ | 0.3 | 0.8 | 0.4 | 2.7 |
| $\Phi^D_c$ | 0.3 | 0.8 | 0.5 | 0.9 |
| $\Phi^T_u$ | 0.24 | 0.52 | 0.05 | 1.75 |
| $\Phi^T_\theta$ | 0.05 | 0.91 | 0.32 | 4.26 |
| $\Phi^T_q$ | 0.04 | 0.86 | 0.40 | 2.28 |
| $\Phi^T_c$ | 0.05 | 0.85 | 0.33 | 4.04 |
| $\frac{\epsilon}{u_*^2}$ | 0.24 | 0.27 | 0.41 | 0.16 |
| $\frac{N_\theta}{\theta_*^2}$ | 0.08 | 0.24 | 0.22 | 0.08 |
| $\frac{N_q}{q_*^2}$ | 0.08 | 0.25 | 0.18 | 0.32 |
| $\frac{N_c}{c_*^2}$ | 0.08 | 0.26 | 0.21 | 0.11 |

Considering the TKE budget equation, $\Phi^T_u$ is in the same order of magnitude as the $\Phi^M$ and $\Phi^D_u$ for the Cases 1 and 4, showing the importance of this component in the variance budget for the convective and the stable ASL regimes. In Case 2, both $\Phi^T_u$ and $\Phi^D_u$ are one order of magnitude smaller than the values of $\Phi^M$. For the tracers, $\Phi^T_{\chi_i}$ is one order of magnitude smaller than the magnitude of other budget components only for Case 1.

The dimensionless power spectra, based on 15-min segment chosen within the 30-min interval of Cases 1, 2, 3, and 4, are illustrated in Fig. 6, in which the estimated inertial subrange for each component is highlighted. It is observed that the subrange detector methodology satisfactorily identifies a range within the inertial subrange. Although the spectra present almost the same inertial subrange frequency interval for different tracers, this is not the case for the majority of segments used in this study. Therefore, the inertial subrange detector is implemented separately for each tracer.

The dimensionless spectra for the tracers collapsed on the examples for Cases 1, 2, and 3 (shown in Fig. 6a, b, and c, respectively). According to Table 3, the variation of $\Phi^T$ among these tracers are relatively small, as expected in the theory explained in Sect. 3.1, after the demonstration of Eq.





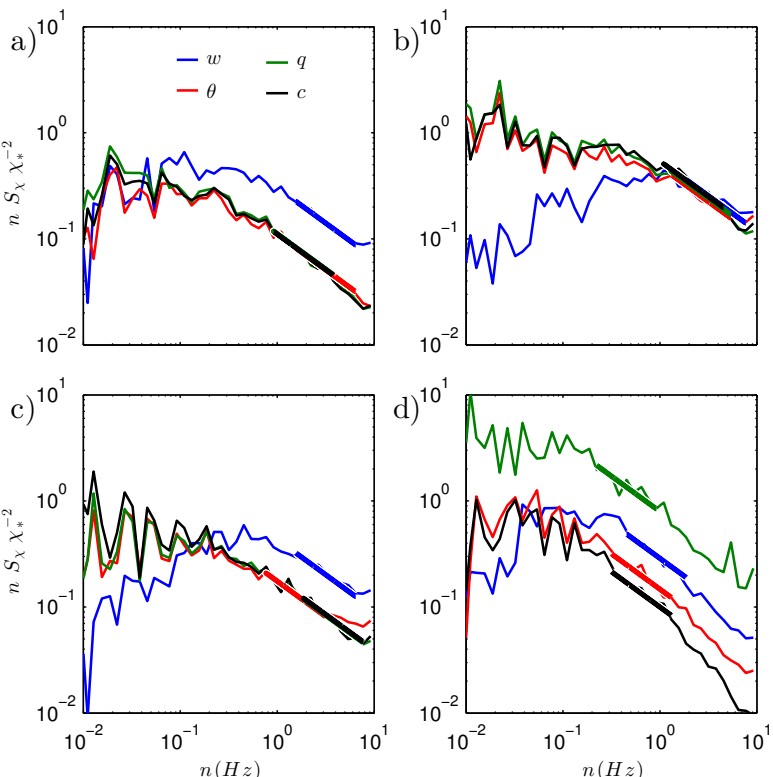

**Figure 6.** Dimensionless power spectra normalized by frequency ($n$ [Hz]) and the respective characteristic scale squared ($\chi_*^{-2}$) for vertical wind velocity (blue), virtual potential temperature (red), specific humidity (green), and specific $CO_2$ content (black). The thick lines indicate the non-linear $-2/3$ power slope fit within the inertial subrange. For each case we used 18,000 data points (15-min segment), collected at 20 Hz sampling rate during the BLLAST campaign at different days on June 2011: (a) 21st, 11 : 37 : 30 UTC; (b) 27th, 00 : 07 : 30 UTC; (c) 18th, 13 : 07 : 30 UTC; and (d) 18th, 22 : 37 : 30 UTC.




(9). In Case 2 however, the transport component for the TKE is significantly different from the ones estimated for the tracers, although the spectra for $w$ also collapsed (Fig. 6$b$). In this case, the ASL stability cannot be considered near-neutral (i.e. $\zeta \approx 0$), which is required in the theory for the cases

in that $\Phi^T$ is invariant also considering the TKE budget equation. Figure 6$d$, taken within the interval of Case 4 is an example where the spectra do not collapse. In this interval, $\Phi^T$ varies considerably among other tracers showing the same order of magnitude as the dynamical production of variance, $\Phi^H$, in the budget equations.

## 5   Summary and outlook

This work brought an overview of how to estimate the dimensionless budget components of the variance equations for TKE, $\theta$, $q$, and $c$ using the EC and ID methods. Important features and peculiarities of each method were discussed in detail.

Among these features, we highlighted that if the transport and mechanical components are invariant to the tracer type, then the dimensionless power spectra normalized by the respective squared

characteristic scale collapse within the inertial subrange.

The results, based on the BLLAST dataset, suggested a set of parameterizations for the transport and dissipation rates of momentum and tracer variances. A linear relationship for the ASL stability range $-1 < \zeta < -0.1$ was proposed for the dimensionless dissipation component of all tracers in the form $\Phi^D_\chi \cong 0.4 + 0.2\zeta$. For the stable ASL regime, although it was observed an increased magnitude

of $\Phi^D$ and $\Phi^T$, also for TKE, the results showed great discrepancies among the different tracers. Thus, we concluded that, in the stable and also near-neutral ASL stability regimes, $\Phi^T_{\chi_i}$ may not be considered identical for different tracers and $\Phi^T_u$ shall not be neglected.

Furthermore, the magnitudes of the characteristic scales of $\theta_*$, $q_*$, and $c_*$ are correlated to $\zeta$, although the sign was not always the same. The friction velocity and the dissipation rate of TKE,

however, were observed to be more sensitive to the mean wind velocity than to the ASL stability regime. Therefore, we suggested that the parameterizations for $\Phi^D_u$ should include more mechanisms rather than only the ASL stability regime.

The described method for estimating each term of the turbulent fluxes and variances budget equations within the ASL has a wide range of applications. Two of them, however, deserve special atten-

tion: $(i)$ for regions where the surface displacement length $(d)$ is large, such as urban areas; and $(ii)$ for data collected from moving platforms, such as buoys and ships.

Under the influence of canopy effects, the measurement height $z$ has to be replaced by $z + d$, and therefore, it is crucial to appropriately determine $d$ (Grimmond et al., 1998; Kanda et al., 2002; Gao and Bian, 2004). While several methodologies with their advantages and disadvantages have been

suggested, some iterative methods (Kanda et al., 2002) require a parameterization for $\Phi^D$, such as the ones proposed in this study. Alternatively, $d$ can be estimated iteratively by solving Eq. (5). In



this scheme, $\Phi^M$, $\Phi^T$, $\Phi^D$ and $\zeta$ depend only on $d$, because the characteristic scales are estimated from Eqs. (12) and (13), the third-order moments in $\Phi^T$ are estimated from EC method, and the dissipation rate of TKE is calculated using the ID method.

For the sensors mounted on moving platforms, the contaminations induced by the platform motion have to be removed or at least to a large extent reduced before the calculation of fluxes, using motion correction algorithms (Edson et al., 1998; Huang et al., 2013; Miller et al., 2008; Prytherch et al., 2015). Although the inertial subrange may be considered isotropic and therefore invariant to platform motion, the elevation of the spectra at the motion-affected frequency band may induce anisotropy as a

noise. Thus, the whole spectrum is modulated and, as a consequence, the dissipation rates are usually overestimated (Bakhoday Paskyabi et al., 2013). This effect will be further enhanced for the offshore moored buoys where data might be collected within the wave boundary layer, and then additional corrections are required in order to account the wave-induced momentum flux (Bakhoday Paskyabi et al., 2015).

## Appendix A: Inertial subrange detector method


It is crucial in the ID method to accurately resolve the inertial subrange. Therefore, we propose an iterative method to evaluate the variations of the independent constants $\alpha$ and $\beta$ defined in Eqs. (14) and (15) by keeping constant values for the dissipation rates within a frequency interval, $I$. From the definition of the inertial subrange, both the dissipation rates and $\alpha$, or $\beta$, are constant and therefore,

the boundaries of $I$ can be defined evaluating the standard deviation of $\alpha$, or $\beta$, in different regions of the spectra.

     The spectra, calculated from 18,000 data sample collected at 20 Hz frequency rate, are divided into 48 frequency blocks (Table 4), which the number of estimates per block are logarithmically distributed, similar to the ones proposed by Kaimal and Gaynor (1983). For each block centered

frequency, $n^*$, the average power spectra, $\overline{S}(n^*)$, and the dissipation rate of TKE, $\epsilon^*(\overline{S_u}(n^*), n^*)$, are calculated using following equation, derived from the Kolmogorov power law.

$$\epsilon^* = \left( \frac{\overline{S_u}(n^*)}{\alpha} \right)^{\frac{3}{2}} n^{*\frac{5}{2}} \left( \frac{2\pi}{\overline{u}} \right). \tag{A1}$$

The iteration among the blocks consists of a loop starting in the centered frequency block $n^* = 0.0922$ Hz (index 22 in Table 4) and moving the interval $I$ toward the higher frequencies, one block

per iteration, up to $n^* = 4.5244$ Hz (index 44). In this method, we suggest an interval $I$ that includes nine consecutive blocks.

     For each iteration, $\alpha$ is estimated for the nine $n^*$ within $I$, keeping $\epsilon^*$ as a constant equal to the one corresponding to the fifth block from $I$ (i.e. the block in the middle of $I$). Finally, the $\alpha$ standard deviation, $\sigma_\alpha$, are calculated for each $I$.





The inertial subrange is then defined as the $I$ which has the smallest $\sigma_\alpha$ after finishing the iteration. If the smallest $\sigma_\alpha$ is greater than $10\%$ of the true $\alpha$, then we consider that the method cannot define an inertial subrange. The $\epsilon$ for the corresponding 30-min segment is then estimated according to the discussion in Sect. 3.3.

    A similar procedure is done for estimating the $I$ for the different tracers spectra. Since $\epsilon$ is now

known, the following Eq. (A2) is applied in an analogous way as it was described for Eq. (A1), but now estimating $\beta$, instead of $\alpha$:

$$N_{\chi_i}^* = \left( \frac{\overline{S_{\chi_i}}(n^*)}{\beta} \right) \epsilon^{\frac{1}{3}} n^{*\frac{5}{3}} \left( \frac{2\pi}{\overline{u}} \right)^{\frac{2}{3}}. \tag{A2}$$

*Acknowledgements.* The authors thank the technicians at IAG-USP and at GFI-UiB and several colleagues for

assistance, in particular for the valuable comments and support of Luciano P. Pezzi, Leonardo Domingues, Pamela Dominutti, Maria M. Hackerott, Vivian Nascimento, Stefan Keiderling, Line Baserud and Valerie Kumer.

    This work was conducted during a scholarship supported by the International Cooperation Program CAPES/COFECUB at the University of Bergen. Financed by CAPES – Brazilian Federal Agency for Support and Evaluation of

Graduate Education within the Ministry of Education of Brazil.

    The BLLAST field experiment was made possible thanks to the contribution of several institutions and supports: INSU-CNRS (Institut National des Sciences de l'Univers, Centre national de la Recherche Scientifique, LEFE-IDAO program), Météo-France, Observatoire Midi-Pyrénées (University of Toulouse), EUFAR (EUropean Facility for Airborne Research) and COST ES0802 (European Cooperation in Science and Technology).

The field experiment would not have occurred without the contribution of all participating European and American research groups, which all have contributed to a significant amount.

    The BLLAST field experiment was hosted by the instrumented site of Centre de Recherches Atmosphériques, Lannemezan, France (Observatoire Midi-Pyrénées, Laboratoire d'Aérologie). BLLAST data are managed by SEDOO, from Observatoire Midi-Pyrénées.

The participation of the Meteorology Group of the Geophysical Institute, University of Bergen was facilitated by contributions of the Geophysical Institute and the Faculty of Mathematics and Natural Sciences under the "smådriftsmidler" scheme, a travel stipend by the Meltzer Foundation in Bergen, and the Short Term Scientific Mission (STSM) scheme within the COST Action ES0802 "Unmanned Aerial Vehicles in Atmospheric Research".




**Table 4.** Spectra logarithmic average smoothing for FFT result originated from 18,000 data sample collected at 20 Hz frequency rate.

| Frequency block index | Number of estimates per block | Centered frequency $n^*$ [Hz] |
|---|---|---|
| 1 | 1 | 0.0011 |
| 2 | 1 | 0.0022 |
| 3 | 1 | 0.0033 |
| 4 | 1 | 0.0044 |
| 5 | 1 | 0.0056 |
| 6 | 1 | 0.0067 |
| 7 | 1 | 0.0078 |
| 8 | 1 | 0.0089 |
| 9 | 1 | 0.0100 |
| 10 | 1 | 0.0111 |
| 11 | 2 | 0.0128 |
| 12 | 3 | 0.0156 |
| 13 | 3 | 0.0189 |
| 14 | 3 | 0.0222 |
| 15 | 5 | 0.0267 |
| 16 | 5 | 0.0322 |
| 17 | 6 | 0.0383 |
| 18 | 7 | 0.0456 |
| 19 | 8 | 0.0539 |
| 20 | 11 | 0.0644 |
| 21 | 12 | 0.0772 |
| 22 | 15 | 0.0922 |
| 23 | 17 | 0.1100 |
| 24 | 21 | 0.1311 |
| 25 | 25 | 0.1567 |
| 26 | 29 | 0.1867 |
| 27 | 36 | 0.2228 |
| 28 | 42 | 0.2661 |
| 29 | 50 | 0.3172 |
| 30 | 61 | 0.3789 |
| 31 | 72 | 0.4528 |
| 32 | 86 | 0.5406 |
| 33 | 102 | 0.6450 |
| 34 | 122 | 0.7694 |
| 35 | 147 | 0.9189 |
| 36 | 174 | 1.0972 |
| 37 | 208 | 1.3094 |
| 38 | 249 | 1.5633 |
| 39 | 296 | 1.8661 |
| 40 | 355 | 2.2278 |
| 41 | 422 | 2.6594 |
| 42 | 505 | 3.1744 |
| 43 | 603 | 3.7900 |
| 44 | 719 | 4.5244 |
| 45 | 859 | 5.40111 |
| 46 | 1025 | 6.44778 |
| 47 | 1224 | 7.69722 |
| 48 | 1461 | 9.18889 |



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
