# Peer review of "Similarity analysis of turbulent transport and dissipation for momentum, temperature, moisture, and $CO_2$ during BLLAST"

_Atmospheric Chemistry and Physics, 2015_

## Author Comment (AC1) · 1 Feb 2016

Line 14 - Where is written "dissipation" should be replaced by "residual". The authors agree that there is a mistake in this phrase that is not in accordance to the conclusions of the manuscript. The correct phrase should be: "In these conditions, the residual component of TKE and tracer variances can have the same magnitude as the other terms in their respective budget equation".

---

## Referee Comment (RC1) · T. Foken (Referee) · 6 Feb 2016

It is a nice and important exercise to check a turbulence data set to determine whether similarity laws of the atmospheric turbulence are fulfilled. This was done by the authors for a data set of the BLLAST experiment. Unfortunately, the theory is not new and is not only the subject of recent textbooks, but even of old ones from the 1960s and 1970s. The presented results are within the usual error bars of the published similarity laws. I am sorry, but I see no necessity to publish results that are not new and that are already well known. I have not seen any special link to the scientific idea of BLLAST, e.g. the modification of similarity relationships in the late afternoon transition with the changing sign of the sensible heat flux and a possible oasis effect. If there is a new

methodological approach that I have overlooked, I propose that this be published in a paper describing the methodology (note). Because I see no possibility for the paper to be published, I am refraining from identifying other problems of the submitted paper.

---

## Author Comment (AC2) · 8 Mar 2016

The authors would like to first of all thank the referee Thomas Foken for his valuable comment and recommendation. Bellow we provide a reply for each statement presented by the reviewer.

**"Unfortunately, the theory is not new and is not only the subject of recent textbooks, but even of old ones from the 1960s and 1970s."**
The Monin-Obukhov Similarity Theory and the Kolmogorov Power Law, used as a background for the work's discussions date back to the beginning and middle of last century as pointed by the reviewer, especially considering the analysis of

momentum and heat budget equations. However, as mentioned in lines 28-32 of the present discussion paper, there is still a need for investigations on these theories particularly for the chemical tracers such as $CO_2$. For example, after an intense bibliography search, we could only find the paper of Ohtaki (1985) and Sørensen and Larsen (2010) that discussed the aspects of the parameterization of $CO_2$ dissipation component as a function of stability, and it was done only for the convective regime. Iwata (2005) emphasizes that the inertial dissipation method has rarely been used to measure $CO_2$ fluxes. After searching in the literature, we also verified that this fact remains true nowadays, since only a few studies are available, among them we can mention Sahlée et al. (2008) and Norman et al. (2012). These recent studies emphasize the need for further investigations, especially for the tracer budget evolution.

**"The presented results are within the usual error bars of the published similarity laws. "**
As mentioned in the discussion paper, the turbulent characteristics of $CO_2$ are very important for climate studies about urban and marine boundary layers. In both applications a range of issues arise, for example platform motion (Prytherch et al., 2015) and horizontal inhomogeneities of cities. These issues decrease the data confidence and increase the error bar of theoretical relations that should be valid in both inland and offshore environments despite their peculiarities. Therefore, it was chosen the confident BLLAST data to perform the analysis. Although we did not provide a deep discussion about the error bars of presented results, we are confident about the accuracy of the presented methodology based on the results of Sørensen and Larsen (2010) as mentioned in Section 3. We are currently working to provide a better error analysis for both spectral and dissipation calculations in order to fill this lack of information.

**"I have not seen any special link to the scientific idea of BLLAST (...) I propose that this be published in a paper describing the methodology. "**
The initial idea of the presented work was to verify the described techniques using a well-established dataset in order to apply it on an upcoming air-sea interaction study. Although BLLAST was idealized mainly for the boundary layer afternoon transition, it fits perfectly for our analysis. Beyond the methodology, our paper supports the scientific hypothesis that the relationship between stability and the dissipation component of the variance budget equation of specific $CO_2$ content is similar to the water vapor and temperature variances. As mentioned before, this hypothesis has not been well described and proofed in the literature yet.

**"I am sorry, but I see no necessity to publish results that are not new and that are already well known."**
We decided to not only evaluate the $CO_2$ but also the momentum and heat fluxes because we would like to provide a better and complete inter-comparison study, following Hill (1989). Nonetheless, as mentioned by the reviewer, it is a nice and important exercise to also check a turbulence data set and confirm previous results. This additional evaluation allowed us to compare the power spectrum of different atmospheric tracers and highlight special features that are still not fully described in the literature, as an example the fact that, in some special conditions, the inertial subrange of different tracers collapse when normalized by the squared characteristic scale, described in Sections 3.1 and 4.3. Thus, our manuscript provides some new aspects on the complicated interactions between different physical and chemical processes which have not been elaborated too much so far.

**"I am refraining from identifying other problems."**
We understand that we may have possible skipped some important references that should be mentioned and make our results unpublishable. Thus we would appreciate if the reviewer could specify or introduce some references, particularly related to the

$CO_2$ dissipation, which may be helpful for the continuation and completing of this work.

Reply bibliography:

Hill, R. J.: Implications of Monin–Obukhov similarity theory for scalar quantities, Journal of the Atmospheric Sciences, 46, 2236–2244, 1989.

Iwata, T., Yoshikawa, K., Higuchi, Y., Yamashita, T., Kato, S., and Ohtaki, E.: The spectral density technique for the determination of CO2 flux over the ocean, Boundary-Layer Meteorology, 117, 511–523, 2005.

Norman, M., Rutgersson, A., Sørensen, L. L., and Sahlée, E.: Methods for estimating air–sea fluxes of CO2 using high-frequency measurements, Boundary-Layer Meteorology, 144, 379–400, 2012.

Ohtaki, E.: On the similarity in atmospheric fluctuations of carbon dioxide, water vapor and temperature over vegetated fields, Boundary-Layer Meteorology, 32, 25–37, 1985.

Prytherch, J., Yelland, M. J., Brooks, I. M., Tupman, D. J., Pascal, R. W., Moat, B. I., and Norris, S. J.: Motion-correlated flow distortion and wave-induced biases in air–sea flux measurements from ships, Atmospheric Chemistry and Physics, 15, 10 619–10 629, 2015.

Sahlée, E., Smedman, A.-S., Rutgersson, A., and Högström, U.: Spectra of CO2 and water vapour in the Marine Atmospheric Surface Layer, Boundary-Layer Meteorology, 126, 279–295, 2008.

Sørensen, L. L. and Larsen, S. E.: Atmosphere–surface fluxes of CO2 using spectral techniques, Boundary-Layer Meteorology, 136, 59–81, 2010.

---

## Referee Comment (RC2) · T. Foken (Referee) · 21 Mar 2016

T. Foken (Referee)

thomas.foken@uni-bayreuth.de

I fully agree with you that the parameterization of similarity functions for carbon dioxide is highly important for climate modelling and I also agree that there are only a very limited number of investigations – not only for carbon dioxide but also for water vapour. But the reason is neither a lack of interest from researchers nor missing data sets (mainly for water vapour), but rather the limitations of our methods or the structure of the atmospheric turbulence in showing any significant difference to the heat exchange. An important developement at the end of the last century was that it became possible to reduce the scatter of the von-Kármán-constant and the turbulent Prandtl number (Högström 1996; Foken 2006). But there was no result available such that the turbulent Schmidt number (relevant for the water vapour exchange, but also for other trace gases) has values that are different to the turbulent Prandtl number. The same was the case for the similarity relationships. There has recently been general agreement to use the turbulent Prandtl number, the universal function for heat, etc., for the water vapour and trace gas exchange as well (Foken 2008). If you want to make a significant contribution to our knowledge, you must show that the turbulent Schmidt number for carbon dioxide and the similarity relationships are different to the normally-used turbulent Prandtl number and similarity relationships within their typical errors (Högström 1988; Högström 1996). The simple presentation of similarity relationships without a discussion of their relevance is less helpful.

References: Foken T (2006) 50 years of the Monin-Obukhov similarity theory. Boundary-Layer Meteorol. 119:431-447. Foken T (2008) Micrometeorology. Springer, Berlin, Heidelberg, 308 pp. Högström U (1988) Non-dimensional wind and temperature profiles in the atmospheric surface layer: A re-evaluation. Boundary-Layer Meteorol. 42:55-78. Högström U (1996) Review of some basic characteristics of the atmospheric surface layer. Boundary-Layer Meteorol. 78:215-246.

---

## Referee Comment (RC3) · Anonymous Referee #2 · 5 Apr 2016

I have asked over 6 additional referees for this manuscript, without any response. Since this paper is outside my expertise, I cannot post my own review. In addition, the one very-critical review is from a thoroughly respected source. I thus can see no more to do, and will have to defer to this review. I can thus see no way of including the manuscript in the special issue.

———————————————